# Development and Optimization of Methylcellulose-Based Nanoemulgel Loaded with *Nigella sativa* Oil for Oral Health Management: Quadratic Model Approach

**DOI:** 10.3390/molecules27061796

**Published:** 2022-03-09

**Authors:** Muhammad H. Sultan, Shamama Javed, Osama Ali Madkhali, Mohammad Intakhab Alam, Yosif Almoshari, Mohammad Ali Bakkari, Durgaramani Sivadasan, Ahmad Salawi, Ameena Jabeen, Waquar Ahsan

**Affiliations:** 1Department of Pharmaceutics, College of Pharmacy, Jazan University, Jazan 45142, Saudi Arabia; sjahmad@jazanu.edu.sa (S.J.); omadkhali@jazanu.edu.sa (O.A.M.); mialam@jazanu.edu.sa (M.I.A.); yalmoshari@jazanu.edu.sa (Y.A.); mbakkari@jazanu.edu.sa (M.A.B.); dsivadasa@jazanu.edu.sa (D.S.); asalawi@jazanu.edu.sa (A.S.); aajabeen@jazanu.edu.sa (A.J.); 2Department of Pharmaceutical Chemistry, College of Pharmacy, Jazan University, Jazan 45142, Saudi Arabia; wmohammad@jazanu.edu.sa

**Keywords:** *Nigella sativa* oil, black seed, dental nanoemulgel, optimization, Box–Behnken statistical design, antimicrobial

## Abstract

The present study aimed to develop a local dental nanoemulgel formulation of *Nigella sativa* oil (NSO) for the treatment of periodontal diseases. NSO purchased from a local market was characterized using a GC–MS technique. A nanoemulsion containing NSO was prepared and incorporated into a methylcellulose gel base to develop the nanoemulgel formulation. The developed formulation was optimized using a Box–Behnken statistical design (quadratic model) with 17 runs. The effects of independent factors, such as water, oil, and polymer concentrations, were studied on two dependent responses, pH and viscosity. The optimized formulation was further evaluated for droplet size, drug release, stability, and antimicrobial efficacy. The developed formulation had a pH of 7.37, viscosity of 2343 cp, and droplet size of 342 ± 36.6 nm. Sustained release of the drug from the gel for up to 8 h was observed, which followed Higuchi release kinetics with non-Fickian diffusion. The developed nanoemulgel formulation showed improved antimicrobial activity compared to the plain NSO. Given the increasing emergence of periodontal diseases and antimicrobial resistance, an effective formulation based on a natural antibacterial agent is warranted as a dental therapeutic agent.

## 1. Introduction

Owing to promising therapeutic effects and reduced side effects, the use of medicinal plants as therapeutic agents is currently enjoying much attention in comparison to modern allopathic medicines [1]. *Nigella sativa* L., of the *Ranunculaceae* family, is an herb with rich historical background [2]. The seeds and oil of *Nigella sativa* L. are used as antihypertensives, liver tonics, diuretics, and antidiarrheal, analgesic, antibacterial, and antifungal agents (3). Thymoquinone (TQ) is the major bioactive component of the oil and is regarded as the main therapeutic constituent of NSO [2,3]. In addition to TQ (30–48%), thymohydroquinone, dithymoquinone, cymene (7–15%), carvacrol (6–12%), 4-terpineol (2–7%), *t*-anethol (1–4%), α-pinene, and thymol are present at good concentrations in NSO. The low toxicity of the oil suggests a wide margin of safety at the therapeutic doses of NSO [4]. Nanoemulgels are a novel drug delivery system candidate that is specifically being explored in the research and development of various topical formulations for both pharmaceutical and cosmeceutical applications. Nanoemulgels consist of a water-in-oil or oil-in-water emulsion embedded in a gel base [5]. A role for *N. sativa* as a dental therapeutic agent in the treatment of periodontal diseases has been established owing to its pharmacological actions, including antibacterial, antifungal, antiplaque, wound-healing, bone-healing, anti-inflammatory, antioxidant, and analgesic activities [6,7,8,9,10]. Periodontal diseases are the top oral health burden in both developing and developed nations; they affect around 20–50% of the population and are most common reason for tooth loss [11]. Several studies reported that antibiotics are often irrationally and overprescribed in dental diseases for both therapeutic and prophylactic reasons, which has led to the rapid emergence of antimicrobial resistance [12]. A few preliminary studies have reported promising results when using NSO in the treatment of periodontal diseases. The implications of NSO in dentistry can be ascertained by detailed preclinical and clinical studies at the cellular and molecular levels to investigate the mechanism of action of NSO in the treatment of periodontal diseases. NSO can be used alone or in suitable combinations with other existing drugs for the effective treatment of infectious dental diseases in a manner that can overcome the problem of drug resistance [13,14,15,16,17]. 

In this study, we aimed to prepare a dental nanoemulgel formulation of NSO with superior properties to the plain NSO and better efficacy in the treatment of periodontitis. A nanoemulgel formulation was selected for development owing to its various advantages over other dental formulations, such as increased patient compliance, ease of application, cost effectiveness, lesser systemic side effects, and ease of local delivery to periodontal pockets. The NSO nanoemulgel formulation is expected to have better biopharmaceutical properties as the nano-sized oil droplets at the target site provide greater surface area for the absorption of NSO. The gel base confers mucoadhesive character to the formulation, allowing it to be present at the target site for a longer time. 

## 2. Materials and Methods

### 2.1. Instruments and Chemicals

Homogenizer (WiseTis Homogenizer, HG—15D, Daihan Scientific Co., Ltd., Seoul, South Korea), pH meter (Mettler Toledo Co. Ltd., Taipei City, Taiwan), Brookfield viscometer (DV-E Viscometer, model no. LWB-111D, Daihan Labtech Co., Ltd., Jakarta, Indonesia), Particle size analyzer Zetasizer (Malvern Panalytical Ltd., Kassel, Germany), Micro pipette (Socorex^®^ Isba SA, Ecublens, Switzerland), UV–Vis spectrophotometer (UV-1800, Shimadzu Corporation, Kyoto, Japan), and GC–MS instrument (Thermo Fisher Scientific, Waltham, MA, USA) equipped with AS 3000 auto sampler, trace ultra GC, and ISQ detector were used in this study. *Nigella sativa* oil (NSO) (>98% pure, lot number: 65550, Safa Al Murooj Mill, Jeddah, Saudi Arabia), methyl cellulose (high viscosity, HiMedia Laboratories Pvt. Ltd., Mumbai, India), polyethylene glycol (PEG) (Sigma Aldrich, St. Louis, MO, USA), and ethanol (96%) (Merck, Darmstadt, Germany) were purchased from authentic suppliers. Ultra-pure double-distilled water was prepared in our laboratory. 

### 2.2. Methods 

#### 2.2.1. Characterization of Plain NSO

Plain NSO was subjected to certain simple laboratory tests to assess its organoleptic and physical properties, such as color, odor, pH, viscosity, density, surface tension, and interfacial tension. 

#### 2.2.2. GC–MS Analysis

The authenticity of NSO with respect to its active constituents was assessed using a GC–MS technique and data were compiled for the percentage availability of its constituents, mainly TQ [18]. The purchased NSO was diluted 100 times using methanol and 2 µL of the diluted solution was injected directly to the GC–MS instrument using autosampler. The carrier gas, helium, was run at a flow rate of 1.2 mL/min throughout the process. Initially, the system was heated to 70 °C, followed by ramping at a rate of 15 °C/min up to 300 °C. The system was held at 300 °C for another 60 min and the fragments thus obtained were detected using an MS detector in electron ionization (EI) mode. The obtained mass spectrum was read and analyzed using Xcalibur software and the constituents were identified by matching their spectra with the spectra of compounds present in the inbuilt databases of the system. Components were quantified using percentage peak area values. 

#### 2.2.3. UV Spectroscopic Analysis

The UV spectroscopic method was adapted from previous study, with slight modifications [19]. Calibration curve was plotted over a range of 50–500 µg/mL for NSO in methanol and the absorbance at 264 nm was measured. 

#### 2.2.4. Formulation Development of NSO Nanoemulgel

High-viscosity methylcellulose E461 was used in this study as a gelling agent. It dissolves in cold liquids to form a clear viscous solution or gel that is non-toxic and non-allergenic in nature. The dental formulation was made in three stages using the methods available in the literature [20,21], with slight modifications. 

##### Formulation of Nanoemulsion

Ultra-pure double-distilled water, NSO, and PEG were mixed in the required proportions (Table 1) and homogenized at a speed of 3000 rpm for 10 min. After homogenization, the mixture was sonicated using probe sonicator for 5–10 min. The developed emulsion was then rested at room temperature for 24 h. 

##### Formulation of Gel Base

In a separate container, the gelling agent methylcellulose was dispersed in cold distilled water and then stirred using an overhead stirrer at a speed of 2000 rpm for 15 min to form a gel base. The prepared gel base was then rested at room temperature for 24 h. 

##### Formulation of Nanoemulgel

The nanoemulsion was gradually added into the gel base and homogenized at a speed of 3000 rpm for 10 min, followed by 3500 rpm for 10 min.

#### 2.2.5. Optimization of the Formulation 

The response surface methodology (RSM) of Box–Behnken statistical design with a quadratic model with 17 runs was used for the optimization of this dental nanoemulgel formulation (DesignExpert 11.0) (Table 2). The effects of formulation factors/variables, such as water (A), oil (B), and gelling agent (C), were observed on the two responses of the formulation, pH (R1) and viscosity (R2), with the help of column, cube (standard error of design), and 3D graphs. ANOVA was applied for the statistical analysis of responses. 

#### 2.2.6. In Vitro Evaluation of NSO Nanoemulgel Formulation

On the basis of optimum pH and viscosity, one final optimized formulation was evaluated for evaluation parameters such as globule size distribution, in vitro drug release percentage, and short-term stability. 

##### Organoleptic Properties

Appearance of the nanoemulgel was noted upon organoleptic examination in terms of color, texture homogeneity, consistency, and phase separation. 

##### Measurement of pH

The pH of dental formulations should correspond to the pH of the saliva (5.5–8.0); therefore, the pH of the preferred nanoemulgel was measured in triplicate and average value was reported. 

##### Viscosity

Rheology plays an important part in determining the flow behavior of such dosage forms. Viscosity of the gelling agent to be used in this study and of the developed nanoemulgel formulations was measured at room temperature (25 °C). The rotational viscometer measured the viscosity when the spindle immersed perpendicularly in the sample rotated at a defined speed ranging from 0.5 to 100 rpm. The corresponding viscosity reading at each speed was noted. 

##### Globule Size Distribution

Globule size distribution was measured by a dynamic light scattering technique using Zetasizer particle size analyzer instrument and the intensity-weighted mean droplet diameter was recorded as the Z-average (nm) [19].

##### In Vitro Drug Release

In vitro drug release from the optimized formulation was evaluated in phosphate buffer pH 6.8 using a dialysis bag method. Typically, 1 g of the NSO nanoemulgel was placed in a dialysis bag (MW 72000), which was clipped to secure the ends. This was placed in a beaker containing 100 mL of phosphate buffer (pH 6.8) maintained at 37 ± 0.5 °C with stirring at 100 rpm. After predetermined time intervals (0, 0.5, 1, 2, 3, 4, 5, 6, 7, 8 h), 2 mL aliquots were withdrawn and replaced with fresh drug-free medium. The amount of NSO present in the samples was analyzed spectrophotometrically by measuring absorbance at 264 nm. The release data were fitted to various release models to understand the release behavior of the nanoemulgel [22]. 

##### Short-Term Stability Studies

Short-term stability studies were conducted on the optimized nanoemulgel by keeping the formulation at various storage conditions—low temperature (4 ± 2 °C), room temperature (25 ± 2 °C), and high temperature (40 ± 2 °C)—for a period of 4 weeks and observing the changes in organoleptic properties and pH [20]. To assess the effect of different temperature conditions on the viscosity of the optimized formulation, the heating–cooling method was adopted under accelerated conditions for seven cycles as reported in the literature [23]. The developed nanoemulgel (n = 3) and the control formulation (n = 3) were stored in refrigerator temperature (4 ± 2 °C), room temperature (25 ± 2 °C), and high-temperature (40 ± 2 °C) conditions. The initial and final viscosities of the formulations were measured.

##### Antimicrobial Efficacy Test of the NSO and Optimized NSO Emulgel Formulation

The minimum inhibitory concentration (MIC) values of the plain NSO sample were determined using the broth dilution assay on *Staphylococcus aureus* strain ATCC 29213. Various concentrations (0.25–256 µg/mL) of the NSO were prepared in DMSO and added to 96-well microtiter plastic plates containing Mueller–Hinton broth medium. Wells were then inoculated with 100 µL of culture containing approximately 5 × 10^5^ colony forming units (CFU)/mL of the test microorganism. Plates were then incubated at 35 ± 2 °C for 24 h. After incubation, plates were examined spectrophotometrically and the turbidity of the medium was measured for each well. The MIC was determined to be the minimum concentration at which inhibition of visible bacterial growth took place. Experiments were performed in triplicate.

The results of the agar well diffusion method were compared to assess the antibacterial activity of plain NSO and the optimized NSO nanoemulgel formulation using *S. aureus* as test organism. Plain NSO (5 µL) dissolved in 1 mL of DMSO, and 0.5 g nanoemulgel formulation dissolved in 1 mL DMSO were incorporated into different wells. The spectrum of antimicrobial activity of plain NSO (positive control) was compared with the prepared nanoemulgel (test sample) in the presence of vehicle control (1 mL DMSO). All the experiments were performed in triplicate (n = 3) and the mean ± standard deviation (S.D.) of diameters of zones of inhibition were measured.

#### 2.2.7. Statistical Analysis

All the experiments were performed in triplicate and the data were expressed as mean ± standard deviation (SD). The statistical analysis of differences was performed using analysis of variance (ANOVA) followed by Student’s *t*-test using GraphPad Prism (version 6.0) software, San Diego, CA, USA, (www.graphpad.com) and *p* values less than 0.05 were considered as statistically significant.

## 3. Results and Discussion

### 3.1. Characterization of NSO

The procured oil was examined for its authenticity using both organoleptic and analytical methods. Physicochemical parameters, such as color, odor, pH, viscosity, surface tension, interfacial tension, and density, were determined. The NSO was observed to be a brownish orange-colored liquid with a specific odor, pH of 6.67, density of 0.91 g/mL, and viscosity of 937.3 cp. The surface tension of the liquid was calculated to be 34.12 dynes/cm and the interfacial tension with respect to distilled water was measured to be 20.20 dynes/cm. The GC–MS analysis of the oil confirmed the presence of TQ was up to 21.18%. The major constituents of the NSO with area values of more than 1% are listed in Table 3. As is evident from the table, all the important constituents of NSO were present in the sample at appropriate concentrations. For the in vitro drug release study, a UV spectroscopic method was employed, and the absorbance was measured at λ_max_ = 264 nm.

### 3.2. Rheological Studies of the Neat Polymer

Mucoadhesive polymers may exhibit the desirable features of a prolonged residence time at the site of drug absorption owing to the increased contact time with the absorbing mucosa. This results in a steep concentration gradient, which favors drug absorption and localization in specified regions to improve and enhance the bioavailability of the drug [24]. Being mucoadhesive, non-toxic, and inert in nature, methylcellulose was selected as a gelling agent in this study. The rheological behavior of the various polymer concentrations was assessed and pseudoplastic behavior was observed for all the tested concentrations of methylcellulose (1, 3, and 5%) (Figure 1).

### 3.3. Optimization of the Developed Nanoemulgel Formulation

Emulgels have emerged as a novel drug delivery system and are currently extensively explored in dentistry [25]. The development of nanoemulgels, a mixture of nanoemulsion and gelling agent, is a relatively more recent approach for a local drug delivery system with good advantages and the potential to be widely used as dental therapeutics [26]. Oil-in-water (*O*/*W*) or water-in-oil (*W*/*O*) nanoemulsions prepared from selected antibacterial oils using a suitable emulsifier can be incorporated into dental nanoemulgels. Emulgels offer various advantages, such as the incorporation of hydrophobic drugs, better loading capacity, better stability, and low cost of production [27]. Dental therapy with local delivery of natural antibacterial agents has recently resulted in commendable clinical outcomes [28,29].

The NSO dental nanoemulgel was prepared and the effects of variables were studied on pH and viscosity using a statistical design for optimization, the Box–Behnken design. Design of experiments (DOE) is a highly systematic approach in finding the most optimized product with the desired characteristics. Formulation factors can be varied in order to get a typical quality of the product. Here, the effects of independent factors—A: Water, B: Oil, and C: Gelling agent—were studied on responses, R1 (pH) and R2 (viscosity); the results are summarized in Table 4. The most optimized formulation obtained after 17 runs was further characterized for appearance, globule size distribution, in vitro release properties, and stability profile. PEG 200 surfactant was chosen as an emulsifying agent, which can be used in the concentration of 0.1% to 10% by weight in nanoformulations [30,31]. Here, PEG 200 concentration was kept constant at 5% by weight for all the prepared formulations.

### 3.4. Effects of Formulation Factors (A, B, C) on the Responses (R1 and R2)

For screening of the factors (A, B, and C), Box–Behnken statistical design was employed and the effect of the concentration of water, oil, and polymer on pH and viscosity was observed. The responses were analyzed with the help of model graphs such as column graphs, cube (standard error of deviation), and 3D graphs. Analysis of variance (ANOVA) was applied to both the responses for their assessment as significant or non-significant in nature. Responses were analyzed individually for better understanding of the effects of factors on the responses.

#### 3.4.1. Analysis of Response R1: pH

The pH of the formulation plays an important role in oral/dental formulations, as acidic or alkaline pH may cause irritation to the oral mucosa. It was important to determine whether the pH of the developed nanoemulgel had a pH close to 7 and if this was maintained during the retention and drug-release periods as the pH of saliva is 6.8. In acidic pH environments (<5.5), teeth begin to demineralize and the risk of cavity formation increases. Teeth remain mineralized when the pH of mouth remains at 6.7 or above, up to 7.5. Acidic formulations with a pH below the critical value can cause erosion of root dentin and the enamel layer [32]. For the estimation of pH, a specific quantity of nanoemulgel was taken and diluted with double-distilled water, and the pH was noted by dipping a glass electrode into the emulgel solution and allowing it to equilibrate for 1 min. 3D surface plots were generated to assess the variation in responses as a function of factors AB, AC, and BC (Figure 2A). In order to keep the pH close to neutral, the formulation with A: Water (10%), B: Oil (5%), and C: Gelling agent (3%) was finalized with an optimum pH of 7.37. From the statistical quadratic model, the coded equation obtained for the response R1 (pH) was:pH = +7.37 + 0.4313 × *A* + 0.0188 × *B* + 0.5000 × *C* − 0.2625 × *AB* − 0.0250 × *AC* + 0.4750 × *BC* − 0.4038 × *A*^2^ − 0.1538 × *B*^2^ − 0.6912 × *C*^2^. (1)

#### 3.4.2. Analysis of Response R2: Viscosity

The rapid elimination of drugs due to the flushing action of saliva is the major issue associated with local therapy of the oral cavity [33]. The treatment of periodontal diseases thus requires a high concentration of the therapeutic agent to be delivered in the periodontal pockets for a prolonged period. Despite some success of local carriers, their application is still beset by insufficient maintenance of drug concentration due to rapid initial release through these methods. The prolonged contact time of the drug with tissues using bioadhesive polymers can significantly improve performance. This local delivery presents numerous advantages over systemic administration in dramatically increasing and sustaining the drug concentration in gingival crevicular fluid, as well as reducing the undesirable systemic side effects [34]. Therefore, a local mucoadhesive system has great potential for the development of localized periodontal therapy. 3D surface plots were generated to assess the variation in responses as a function of factors AB, AC, and BC (Figure 2B). The viscosity of the formulation increased considerably with an increase in the polymer concentration. The formulation with A: Water (10%), B: Oil (5%) and C: Gelling agent (3%) was finalized as optimum, with a viscosity of 2343 cp. From the statistical quadratic model, the coded equation obtained for the response R2 (viscosity) was:
Viscosity = +2343.00 − 17.37 × *A* − 3.25 × *B* + 862.13 × *C* + 3.25 × *AB* + 118.50 × *AC* − 43.25 × *BC* − 25.00 × *A*^2^ − 602.75 × *B*^2^ + 13.00 × *C*^2^.(2)

### 3.5. Characterization of Final Optimized Formulation

On the basis of the analysis of responses R1 (pH 7.37) and R2 (viscosity 2343 cp), the formulation coded with A: Water (0), B: Oil (0) and C: Gelling agent (0) was selected for further evaluation parameters. The final formulation was creamy white in appearance and exhibited pseudoplastic flow. Globule size distribution, the in vitro drug-release profile, short-term stability studies were performed on the final optimized dental nanoemulgel.

#### 3.5.1. Globule Size Distribution

The nanodispersion of oil and water was thermodynamically stabilized by the interfacial layer of the surfactant polyethylene glycol. Dynamic light scattering measures the fluctuation in the intensity of scattering by droplets due to Brownian motion. From the droplet size distribution, it was seen that homogenization of the formulation followed by sonication technique produced smaller droplet size, as the droplet size reduced from 712 nm in homogenization only to 342 nm in homogenization followed by sonication, as shown in Figure 3A,B. The larger globule size achieved with the homogenization technique only can be a result of improper emulsification. In contrast, the emulsification was improved by the application of sonication and a reduction in the droplet size was observed.

#### 3.5.2. In Vitro Release Studies

It is important for a dental formulation to show optimum mucoadhesion so that it remains adhered to the dental mucosa. The formulation should release the drug over a prolonged period of time; therefore, in vitro release is also considered to be an important evaluation parameter. The in vitro release study of the optimized formulation gave insight into the drug-release profile and efficiency of the drug delivery system to achieve the sustained release of NSO. The formulation showed good ability to retain the drug for up to 8 h (Figure 4), confirming that the developed nanoemulgel can be used in the therapeutic arsenal of periodontal therapy where it is needed to deliver the drug locally into the periodontal pockets for a long period of time. Methylcellulose plays a very important role in sustaining the drug delivery owing to its mucoadhesive character. The release data were then fitted to various release kinetics models, and a linear relationship was observed when the fraction drug release was plotted against the square root of time, with the highest correlation coefficient (R^2^) of 0.995, showing that the release followed the Higuchi model (Figure 5). The Higuchi model in conjunction with the Korsmeyers–Peppas model suggested a non-Fickian diffusion release (n = 0.63) from the gel matrix.

### 3.6. Stability Studies

To assess the short-term stability, the nanoemulgel formulations were kept in suitable containers and were stored at low temperature (4 ± 2 °C), room temperature (25 ± 2 °C), and high temperature (40 ± 2 °C) conditions for a period of 4 weeks. At the end of the storage period, formulations were observed for physical appearance and change in pH (Figure 6). Stability study results revealed that the nanoemulgel was stable at room temperature for a period of 4 weeks; the pH changed from 7.37 to 6.93 during the storage period, suggesting that the selection of excipients and their concentrations was appropriate. At low temperature, the formulation also remained stable in terms of acidity, as the pH changed from 7.37 to 7.11; however, at high temperature, the formulation showed instability as the pH changed considerably and became acidic (pH = 6.57). This could be due to the instability of polymer when exposed to higher temperature for 4 weeks. The physical appearance and organoleptic properties of formulations kept at different temperature conditions after 4 weeks were observed to be similar and no marked differences were observed.

The thermodynamic stability of the formulation was tested under the conditions of temperature stress using a heating–cooling accelerated method. Seven cycles between the three temperature conditions were applied to the optimized and control formulations. The viscosities of formulations were measured before the study and after the stress conditions, as presented in Figure 7. As evident from the figure, no significant effect of the heating–cooling cycles was observed on the viscosity of both the control and optimized formulations. The final viscosity of the control formulation changed from its initial value of 2575 ± 55 cp to 2540 ± 51 cp. Similarly, the viscosity of the optimized nanoemulgel changed from 2343 ± 27 cp before the study to 2325 ± 18 cp after the study. This showed that the developed nanoemulgel formulation was stable under temperature stress conditions.

### 3.7. Antibacterial Susceptibility Test against S. aureus

The results of the MIC assay showed excellent antimicrobial efficacy of the tested NSO as the MIC value was found to be 1.0 µg/mL against the tested *S. aureus* strain. The comparative antibacterial susceptibility test carried out between the plain NSO and optimized NSO-nanoemulgel formulation revealed that NSO, once incorporated into the nanoformulation, showed better antibacterial activity as compared to the plain oil and the vehicle control for a period of 48 h. The enhanced activity of NSO-nanoemulgel could be attributed to the presence of oil in the form of nanosized globules in the formulation (Figure 8). These results were concordant with a previous study, where the NSO showed superior activities against the Gram positive cocci [35].

The oral cavity has a complex environment as a myriad of diverse microorganisms exists in resistant biofilms. It consists of diverse anatomical structures and is constantly flushed by saliva due to the high flow rate of gingival crevicular fluid (20 µL/h). Dental diseases are considered as a major health problem throughout the world and their long-term treatment is still warranted [36]. To overcome this problem, there are many synthetic drugs available in the market, but these are associated with numerous side effects and antimicrobial resistance that prohibit their long-term use. Natural remedies are more acceptable as they are considered to be safer, with fewer side effects than synthetic remedies [1]. Herbal drugs cannot be used as such for the treatment of oral diseases and need to be developed into a suitable dosage form. In the case of dental delivery, a toothpaste, mouthwash, strip, chips, gels (hydrogels, nanoemulgels), microparticles, or nanoparticles formulations are suitable [37]. The local delivery of therapeutic agents to the dental pockets offers loco-regional and targeted sites of action, enhanced therapeutic efficacy, as well as reduced side effects that are associated with systemic drug delivery. The time has come to develop cost-effective formulations for the treatment of periodontal diseases in order to alleviate the financial burden on the global community. Odontogenic infections that originate in the teeth or closely surrounding structures of the teeth, such as dental caries, gingivitis, and periodontitis, require special attention.

## 4. Conclusions

The efficient delivery of hydrophobic and poorly water-soluble drugs has been a challenging issue, along with stability and bioavailability. Nanoemulgels have been proven to be a standout candidate for their drug delivery, as they possess nanoscaled drug emulsion in a gel base. The developed NSO nanoemulgel showed good potential for the treatment of periodontal diseases owing to the propitious and practical features for the topical delivery of NSO. The incorporation of NSO into a nanoemulgel formulation will help to increase patient compliance through ease of application, along with improved efficacy. Cost-effectiveness and increased mucoadhesiveness are the other associated advantages, making the nanoemulgel an attractive approach compared with other conventional topical formulations. The nanosized NSO droplets are expected to help in maintaining closer mucosal contact, thereby offering greater surface area for the penetration of NSO and ensuring improved drug concentration at the target site. However, this study had a few limitations; namely more sophisticated characterization techniques should be employed and the efficacy of the developed formulation should be assessed and compared using in vivo experiments. The effects of various other emulsifiers and gelling agents can also be studied on the globule size, stability, drug release, viscosity, and pH of the formulation. Moreover, NSO can be combined with other natural or synthetic antimicrobial agents and the combination can be used to develop nanoemulgel formulations for preclinical and clinical testing. Nevertheless, more preclinical and clinical studies are warranted to establish the usefulness of this formulation for the treatment of periodontal diseases.

## Figures and Tables

**Figure 1 molecules-27-01796-f001:**
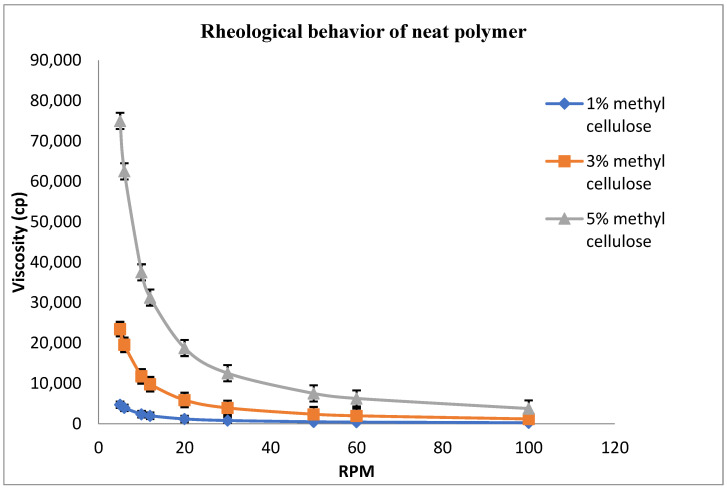
Rheological behavior of the neat polymer solutions (n = 3).

**Figure 2 molecules-27-01796-f002:**
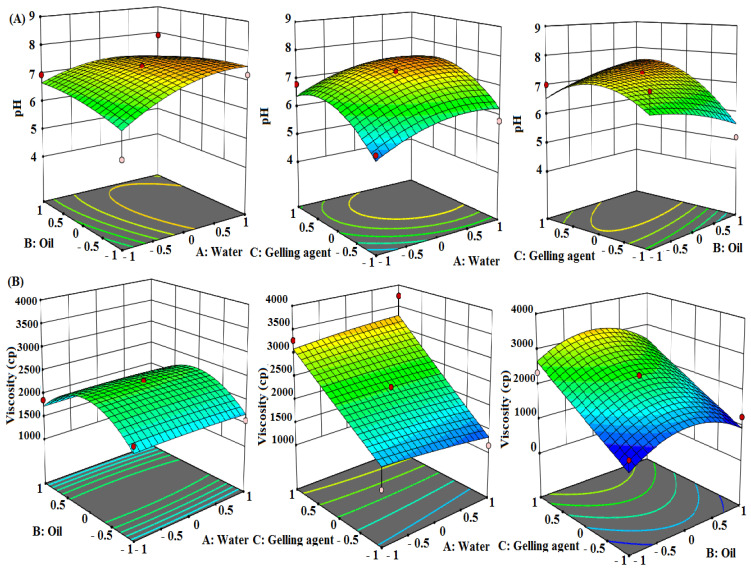
3D surface plots between the factors AB, AC, and BC, showing the effects of oil, water, and polymer concentration on (**A**) pH and (**B**) viscosity.

**Figure 3 molecules-27-01796-f003:**
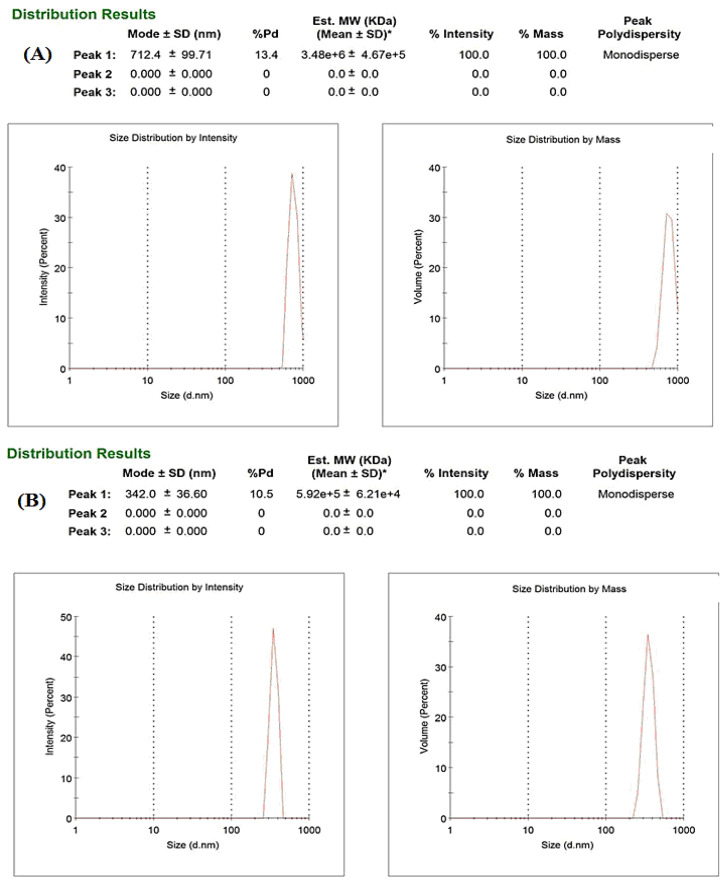
Globule size distribution after (**A**) homogenization only and (**B**) homogenization followed by sonication.

**Figure 4 molecules-27-01796-f004:**
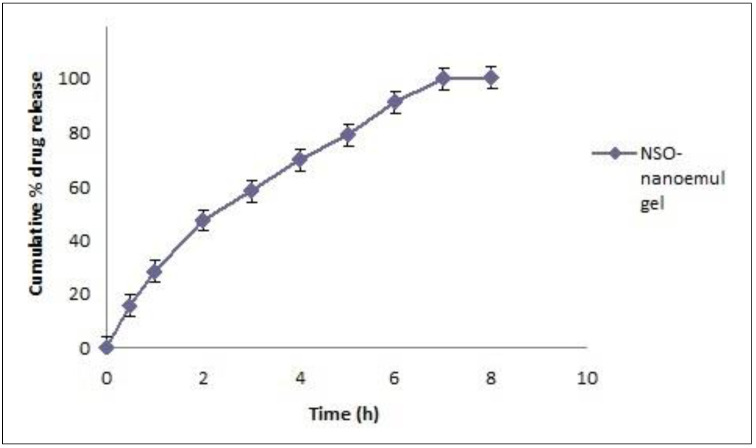
Drug release plot for the developed NSO nanoemulgel showing sustained drug release up to 8 h (n = 3).

**Figure 5 molecules-27-01796-f005:**
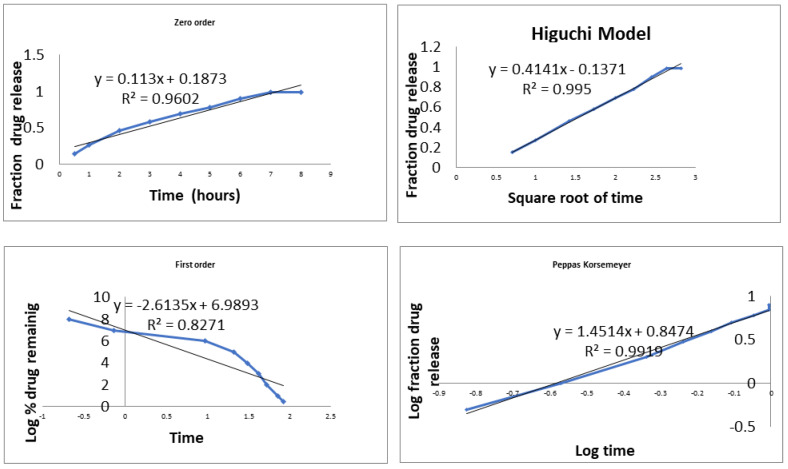
Drug release kinetic plots obtained for the developed NSO nanoemulgel showing R^2^ values for various kinetic models.

**Figure 6 molecules-27-01796-f006:**
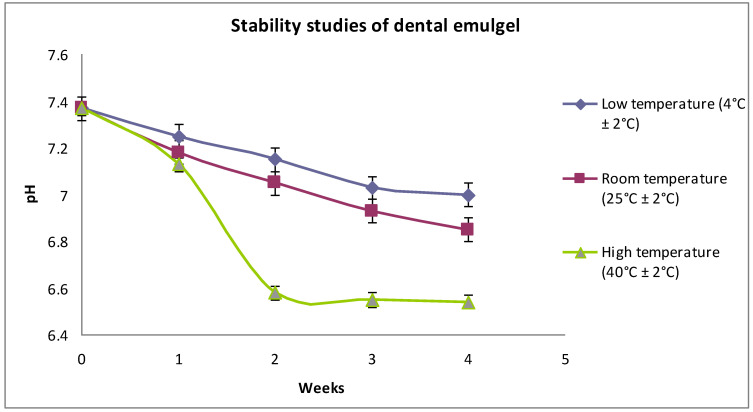
Effect of storage temperature on the short-term stability of NSO nanoemulgel (n = 3).

**Figure 7 molecules-27-01796-f007:**
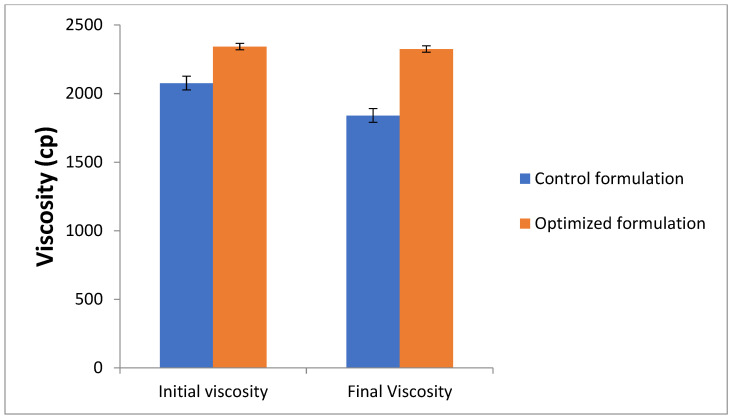
Effect of seven cycles of heating–cooling conditions on viscosity of control and optimized formulations.

**Figure 8 molecules-27-01796-f008:**
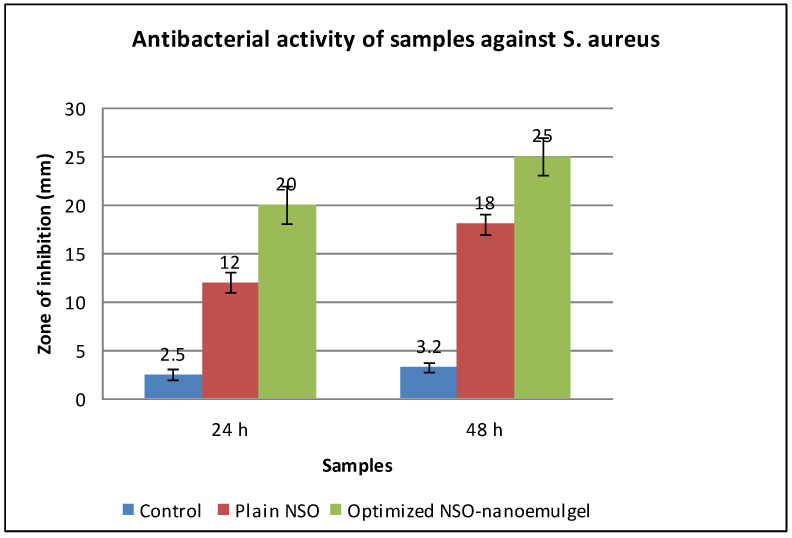
Antibacterial activity of control, plain NSO, and NSO-nanoemulgel formulation against *S. aureus* as the test microorganism (n = 3).

**Table 1 molecules-27-01796-t001:** Ingredients used in NSO nanoemulgel formulation development.

Components	Ingredients	Concentration Range (% *w*/*w*)
Oil phase	*Nigella sativa* oil	0–10%
Water phase	Double-distilled water	5–15%
Emulsifying agent	Polyethylene glycol	5%
Gelling agent	Methylcellulose	1–5%
Total weight of the formulation		20 g

**Table 2 molecules-27-01796-t002:** Optimization table showing factors with their levels (−1, 0, and +1).

Factors	Levels		
	−1	0	+1
A: Water	5%	10%	15%
B: Oil	0%	5%	10%
C: Gelling agent	1%	3%	5%

**Table 3 molecules-27-01796-t003:** List of major constituents identified in the NSO sample.

S. No	Name of the Constituent	Area (%)
1.	*p*-Cymene	26.30%
2.	Thymoquinone	21.18%
3.	Ledol	10.96%
4.	9,12-octadecadienoic acid (Z,Z)-	9.09%
5.	Nerol, methyl ether	2.50%
6.	Lavandulol, methyl ether	2.50%
7.	1,4- Methanoazylene decahydro-4,8,8-trimethyl-9-methylene	2.36%
8.	Carvacrol	2.12%
9.	α-Pinene	1.74%
10.	Squalene	1.53%
11.	Hydroquinine	1.21%
12.	Androst-4-ene 3,11,17-trione	1.14%

**Table 4 molecules-27-01796-t004:** Coded and uncoded level of the factors with their effects on responses (R1 and R2).

Run	Factor 1 A: Water	Factor 2 B: Oil	Factor 3 C: Gelling Agent	Response 1 R1: pH	Response 2 R2: Viscosity (cp)
1	−1	1	0	6.95	1862
2	0	−1	1	7	2343
3	0	1	−1	5.1	1250
4	1	0	1	7	3749
5	0	0	0	7.37	2343
6	−1	−1	0	5.2	1875
7	1	1	0	7.9	1562
8	0	0	0	7.37	2343
9	−1	0	1	6.8	3275
10	0	1	1	6.8	2250
11	0	0	0	7.37	2343
12	−1	0	−1	5.5	1150
13	1	0	−1	5.8	1150
14	0	0	0	7.37	2343
15	1	−1	0	7.2	1562
16	0	−1	−1	7.2	1170
17	0	0	0	7.37	2343

## Data Availability

Not applicable.

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
