# Peer review of "Development and Optimization of Methylcellulose-Based Nanoemulgel Loaded with Nigella sativa Oil for Oral Health Management: Quadratic Model Approach"

_molecules, 2022, doi:10.3390/molecules27061796_

Round 1

Reviewer 1 Report

1-It is not common to mention the equipment in MATERIALS AND METHODS section.

2-In section 2.2.3.2. Formulation of gel base, it was mentioned that, “gelling agent methylcellulose was dispersed in cold distilled  water followed by homogenization using a homogenizer at a speed of 2000 rpm for 15  min to form a gel base”. It is not common to prepare the gel using homogenizer, is there reference to such method?

3-As mentioned in section, 3.4.1 “Increase in the concentrations of oil and polymer resulted in a decrease in the pH of the developed  nanoemulgel.” However, the equation is contradictory to your explanation. “pH = +7.37+0.4313*A+ 0.0188*B + 0.5000*C”

4-As mentioned in section, 3.4.2 “The viscosity of the formulation increased considerably with increase in the polymer concentration.” However, the equation is contradictory to your explanation. “Viscosity = +2343.00-17.37*A-3.25*B+862.13*C

5-Statistical analysis method is missing in the manuscript.

6- As mentioned in the title Quadratic model approach was applied, however, anova of the data and fit statistics are not shown as a proof that Quadratic model was obtained successfully.

7- In stability study, particle size, in vitro release and viscosity were not evaluated?

Author Response

I want to express my sincere thanks to you and the reviewers for the positive feedback, time and effort in providing constructive comments for improving our manuscript. Based on those comments, we have revised the manuscript thrugh clarifing, updating, and restructuring the text for flow and logic and having the resulting text edited by a pharmacist who natively speaks english.

It is my great hope that the manuscript is now fully suitable for publication in your esteemed journal.

Reviewer 2 Report

The authors aimed to evaluate the effects of a dental nanoemulgel formulation of Nigella  sativa oil (NSO) for the treatment of periodontitis. Nanoemulgel formulation was opted to develop owing to its various advantages over other dental formulations such as increased patient compliance, ease of application, cost effectiveness, lesser systemic side effects and local delivery to periodontal pockets.

The study covers some issues that have been overlooked in other similar topics. The structure of the manuscript appears adequate and well divided in the sections. Overall, the manuscript was written in good English and easy to understand and follow. Some of the comments that would improve the overall quality of the study are:

1-) Please also check typos thorough the text

2-) Conclusion Section: This paragraph required a general revision to eliminate redundant sentences and to add some "take-home message".

Author Response

(The authors gave the same response as above.)

Reviewer 3 Report

The article is well organized, the materials and methods allow replication of the methodology and the results are well presented.

I only have a few suggestions:

I would add a null hypothesis at the end of the introduction section.

There should be a paragraph on the limitations of this study and on the needed future research.

I would add some more information on the clinical implication of this findings, for both clinicians and patients, as well as on future clinical research needed to validate these results.

I would try to add some up-to-date references. Most of the references were published over 5 years ago.

Author Response

(The authors gave the same response as above.)
